



# Simultaneous measurements of particle number size distributions at ground level and 260 m on a meteorological tower in urban Beijing, China

Wei Du[1,2], Jian Zhao[1,2], Yuying Wang[3], Yingjie Zhang[1,4], Qingqing Wang[1], Weiqi Xu[1,2], Chen Chen[1], Tingting Han[1,2], Fang Zhang[3], Zhanqing Li[3], Pingqing Fu[1,5], Jie Li[1], Zifa Wang[1,5], Yele Sun[1,5*]

[1]State Key Laboratory of Atmospheric Boundary Layer Physics and Atmospheric Chemistry, Institute of Atmospheric Physics, Chinese Academy of Sciences, Beijing 100029, China

[2]College of Earth Sciences, University of Chinese Academy of Sciences, Beijing 100049, China

[3]College of Global Change and Earth System Science, Beijing Normal University, Beijing 100875, China

[4]School of Atmospheric Physics, Nanjing University of Information Science and Technology, Nanjing 210044, China

[5]Center for Excellence in Regional Atmospheric Environment, Institute of Urban Environment, Chinese Academy of Sciences, Xiamen 361021, China

*Correspondence to: Y. L. Sun (sunyele@mail.iap.ac.cn)

**Abstract.**  Despite extensive studies into characterization of particle number size distributions at ground level, real-time measurements above the urban canopy in the megacity of Beijing has never been performed to date. Here we conducted the first simultaneous measurements of size-resolved particle number concentrations at ground level and 260 m in urban Beijing from 22 August to 30 September. Our results showed overall similar temporal variations in number size distributions between ground level and 260 m, yet periods with significant differences were also observed. Particularly, accumulation mode particles were

highly correlated ($r^2$ = 0.85) at the two heights while Aitken mode particles presented more differences. Detailed analysis suggests that the vertical differences in number concentrations strongly depended on particle size, and particles with mobility diameter between 100 – 200 nm generally showed higher concentrations at higher altitudes. Particle growth rates and condensation sinks were also calculated which were 3.2 and 3.6 nm h$^{-1}$, and 2.8×10$^{-2}$ and 2.9×10$^{-2}$ s$^{-1}$, at ground level and 260 m, respectively. By linking particle growth with aerosol composition, we found that organics appeared to play an important role

in the early stage of the growth (9:00 – 12:00) while sulfate was also important during the later period. Positive matrix factorization of size-resolved number concentrations identified three common sources at ground level and 260 m including a factor associated with new particle formation and growth events (NPE), and two secondary factors that represent photochemical processing and regional transport, respectively. Cooking emission was found to have a large contribution to small particles, and showed much higher concentration at ground level than 260 m at dinner time. This result has significant implications that





investigation of NPE at ground level in megacities needs to consider the influences of local cooking emissions. The impacts of

regional emission controls on particle number concentrations were also illustrated. Our results showed that regional emission

controls have a dominant impact on accumulation mode particles by decreasing gas precursors and particulate matter loadings,

and hence suppressing particle growth. In contrast, the influences on Aitken particles were much smaller due to the enhanced

new particle formation (NPF) events.

**1 Introduction**

With frequent occurrence of haze episodes, the megacity of Beijing is facing with severe air pollution problems as

indicated by high concentrations of ambient aerosol particles. For example, the annual average concentration of $PM_{2.5}$ was 80.6

$\mu g\ m^{-3}$ in 2015, which is more than twice the China National Ambient Air Quality Standard (35 $\mu g\ m^{-3}$ as an annual average)

(http://www.bjepb.gov.cn/bjepb/413526/413663/413717/413719/index.html). Fine particles can reduce atmospheric visibility

significantly, exert harmful effects on public health, and even have potential impacts on regional and global climate. As a result,

extensive efforts have been devoted to characterize the sources, formation mechanisms, and evolution processes of aerosol

particles in recent years (Ma et al., 2012;Takegawa et al., 2009;Sun et al., 2010;Sun et al., 2014;Sun et al., 2015). Among these

studies, particle number concentrations are one of the greatest concerns because particles can rapidly grow from a few

nanometers to tens and even hundreds of nanometers in a short time, and hence play a significant role in haze formation (Guo et

al., 2014). However, our understanding of the formation and growth of aerosol particles is not complete, particularly in highly

polluted environments (Kulmala et al., 2016).

In the past decades, extensive studies have been conducted to characterize particle number size distributions in Beijing at

ground level (Wehner et al., 2004;Yue et al., 2009;Wu et al., 2011;Gao et al., 2012;Wang et al., 2013b). The first continuous

measurements of aerosol number size distributions within the city area of Beijing indicated a high variability in number

concentrations, and the variations were substantially different among dust storm, clean and polluted periods (Wehner et al.,

2004). Yue et al. (2009) also found a clear shift of maximum diameter from 60 nm in clean days to 80 nm during polluted days.

Most of previous studies were focused on new particle formation and growth events (NPE) (Wehner et al., 2004;Wu et al.,

2011;Gao et al., 2012;Yue et al., 2010;Zhang et al., 2011;Wang et al., 2015). While new particle formation events (NPF) are

mostly observed under conditions with low relative humidity and clean air masses (Wu et al., 2007;Wehner et al., 2004),

particle growth events are strongly associated with high relative humidity (Gao et al., 2012). The roles of chemical species in

NPE in Beijing were also explored in several studies. For example, organics were found to be the dominant species in 23 new

particle formation events during the Beijing Olympic Games (Zhang et al., 2011), and likely played a major role in NPF and



growth (Wang et al., 2015) although sulfuric is also important as well (Yue et al., 2009;Yue et al., 2010). However, most of these studies were conducted at ground site which is subject to the influences of multiple local sources, e.g., traffic and cooking

emissions. Indeed, the source apportionment of particle numbers with positive matrix factorization showed significant contributions of traffic emissions and combustion sources to the total number concentration (Wang et al., 2013b;Liu et al., 2014). Therefore, measurements of size-resolved number concentrations at high altitude with less local influences are essential for elucidating the NPF and growth mechanisms, and also the role of regional transport in haze formation.

During this study period, strict emission controls were implemented in Beijing and surrounding regions, e.g. Hebei,

Tianjin, and Shandong, from 20 August to 3 September to ensure the good air quality during the China Victory Day (V-day) Parade on 3 September 2015. The control measures such as restricting the number of vehicles, shutting down factories and power plants, stopping construction activities, and etc. were even stricter than those implemented during the Asia-Pacific Economic Cooperation (APEC) summit in 2014 (Sun et al., 2016). Several studies have addressed the impacts of regional emission controls on aerosol composition and gaseous species (Li et al., 2016;Zhao et al., 2016;Han et al., 2016). The results

are overall consistent showing significant reductions in most aerosol and gaseous species during the control period (CP, 22 August – 3 September). A recent study by comparing the number size distributions with those during the same period in 2010-2013 at a rural site in Beijing illustrated the most reductions in accumulation mode particles and condensation sink (CS) during the V-day period (Shen et al., 2016). Despite this, our understanding of the impacts of emission controls on particle number size distributions is far from complete.

Here, we conducted the first simultaneous measurements of particle number size distributions at two different heights, i.e., ground level and 260 m within the city area of Beijing from 22 August to 30 September. This study is unique by providing an experimental opportunity to investigate the vertical differences and processes of particle number size distributions and also the impacts of regional emission controls. The size-resolved particle number concentrations, diurnal variations, particle growth rates and its relationship with aerosol composition at ground level and 260 m are compared in detail, and the impacts of

emission controls on particle number concentrations in different sizes are elucidated. In addition, the sources of particle numbers at the two different heights are investigated with positive matrix factorization.

## 2    Experimental method

### 2.1 Sampling and data analysis

The sampling site is located at the Tower Branch of Institute of Atmospheric Physics, Chinese Academy of Sciences

between the north third and fourth ring road in Beijing. Two Scanning Mobility Particle Sizers (SMPS) were deployed for





simultaneous measurements of particle number size distributions at ground level and 260 m on the Beijing 325 m

meteorological tower. At 260 m, the size-resolved particle number concentration (15 – 685 nm) was measured *in-situ* by a

Condensation Particle Counter (CPC, TSI, 3775) equipped with a long Differential Mobility Analyzer (DMA, TSI, 3081A). The

time resolution is 5 min. Comparatively, an SMPS as part of an unattended multifunctional Hygroscopicity-Tandem Differential

Mobility Analyzer (H-TDMA) developed by the Guangzhou Institute of Tropical and Marine Meteorology, China

Meteorological Administration (ITMM, CMA) was used to measure particle number concentrations (10 – 400 nm) at ground

level. A detailed description of the H-TDMA was given in Tan et al. (2013).

The non-refractory submicron aerosol (NR-PM$_1$) species, including organics (Org), sulfate (SO$_4$), nitrate (NO$_3$),

ammonium (NH$_4$), and chloride (Chl), were measured at ground level by an Aerodyne High-resolution Time-of-Flight Aerosol

Mass Spectrometer (HR-AMS) and at 260 m by an Aerosol Chemical Speciation Monitor (ACSM), respectively. Co-located

black carbon (BC) was measured by a seven-wavelength (AE33) and a two-wavelength Aethalometer (AE22, Magee Scientific

Corp.) at 260 m and ground level, respectively. The meteorological variables, including wind speed (WS), wind direction (WD),

relative humidity (RH), and temperature (*T*) were obtained from the measurements on the meteorological tower. The operations

of the HR-AMS, ACSM, and Aethalometers and subsequent data analysis are detailed in Zhao et al. (2016). All the data in this

study are reported in Beijing Local Time (= UTC + 8h).

Figure S1 shows a comparison of the total PM$_1$ mass (= NR-PM$_1$ + BC) with that derived from the SMPS measurements at

ground level and 260 m. The particle number concentrations between 15 nm and 400 nm were converted to mass concentrations

using chemically-resolved particle density (Salcedo et al., 2006). As shown in Fig. S1, the time series of PM$_1$ was highly

correlated with that from SMPS measurements at both ground level (r$^2$ = 0.94) and 260 m (r$^2$ = 0.95). We also noticed some

differences in the regression slopes, which are 0.44 and 0.66 at ground site and 260 m, respectively. The reasons are not very

clear yet, but likely due to the different size distributions at the two different heights (Section 3.1)

**2.2 Particle growth rates and condensation sink**

The particle growth rates (GR) at ground level and 260 m were calculated using Eq. (1).

$$GR = \frac{\Delta D_m}{\Delta t} \qquad\qquad (1)$$

Where $D_m$ is the geometric mean diameter from the log-normal fitting of each size distribution, $\Delta D_m$ is the increase in diameter

during the growth period of $\Delta t$.

Condensation sink (CS) indicating how rapidly vapor molecules can condense onto pre-existing aerosols is calculated

using Eq. (2) (Nieminen et al., 2010).



$$CS = 2\pi D \sum_i \beta_{Mi} D_{p,i} N_i \qquad (2)$$

Where $D$ is the diffusion coefficient of the condensing vapor, $D_p$ and $N$ is the particle diameter and the corresponding number

concentration, and $\beta_M$ is the transitional regime correction factor expressed as Eq. (3).

$$\beta_M = (K_n + 1)\bigg/\left(1 + 0.377K_n + \frac{4}{3}\alpha^{-1}K_n^2 + \frac{4}{3}\alpha^{-1}K_n\right) \qquad (3)$$

Where $\alpha$ is assumed to be unity, and $K_n$ is the Knudsen number. It should be noted that the CS calculated on the basis of dry

particle number size distributions might be underestimated since ambient RH was not considered (Reutter et al., 2009).

**2.3 Source apportionment of size-resolved particle number concentrations**

Positive matrix factorization (PMF2.exe, v 4.2) was performed on the size-resolved number concentrations (Ulbrich et al.,

2009;Paatero and Tapper, 1994) to resolve potential sources. In this study, the measurement uncertainties were estimated using

an equation-based approach that was detailed in Ogulei et al. (2007). The required measurement errors ($\sigma_{ij}$) were first calculated

using Eq. (4)

$$\sigma_{ij} = C_1 \times (X_{ij} + \bar{X}_J) \qquad (4)$$

Where $C_1$ is a constant value assumed to be 0.01; $X_{ij}$ is the measured particle number concentration; $\bar{X}_J$ is the arithmetic mean

value for $j^{th}$ size bin. The measurement uncertainties (Unc) were then calculated with Eq. (5)

$$Unc_{ij} = \sigma_{ij} + C_2 \times X_{ij} \qquad (5)$$

Where $\sigma_{ij}$ is the estimated measurement errors and $C_2$ is a constant value assumed to be 0.1. After a careful evaluation of the

PMF results, five and four factors were chosen at ground level and 260 m, respectively. A more detailed diagnostics of PMF

results are presented in Figs. S2 and S3.

### 3 Results and discussion

### 3.1 Characterization of size distribution

The temporal variations of size-resolved number concentrations and aerosol species at ground level and 260 m are shown

in Fig. 1. The size-resolved particle number concentrations showed overall similar evolutionary patterns between ground level

and 260 m, and high number concentrations of large particles were generally associated with correspondingly higher

concentrations of aerosol species, e.g., the periods of case 1 and case 2 in Fig. 1. However, periods with substantially different

number size distributions were also observed. For example, we observed significantly higher particle number concentrations at

ground level than 260 m at evening time on 26 August and 1 September due to the influences of local cooking emissions. On

average, the particle numbers showed a broader size distribution at 260 m than ground level, peaking at approximately 85 and




45 nm, respectively (Fig. 1c). The log-normal distribution fitting further illustrated three size modes at both ground level and 260 m. While the second mode with geometric mean diameter (GMD) peaking at 41 nm accounted for the largest number fraction at ground level (52%), the largest mode (GMD = 116 nm) dominated the total number of particles at 260 m, accounting for 62%. Such differences were likely due to the stronger influences of local sources (e.g., cooking) with higher emissions of

smaller particles, and more influences of regional transport with aged large particles at 260 m.

Figure 2 shows the comparisons of the total number concentrations (15 – 400 nm, $N_{15-400}$) and those for three modes including small Aitken mode (15 – 40 nm, $N_{15-40}$), large Aitken mode (40 – 100 nm, $N_{40-100}$), and Accumulation mode (100 – 400 nm, $N_{100-400}$) between ground level and 260 m. The variation trends of the total number concentrations at the two heights tracked relatively well ($r^2 = 0.40$, slope = 0.71), while the average number concentration from 15 nm to 400 nm at 260 m (7473

$\pm 4324$ cm$^{-3}$) was 26% lower than that (10134 $\pm 4680$ cm$^{-3}$) at ground level. The total particle number concentrations at ground level were generally lower than those previously observed in Beijing mainly due to the smaller size range measured in this study (Wang et al., 2013b;Yue et al., 2013;Yue et al., 2009;Wu et al., 2008). The $N_{15-400}$ ratio of 260 m to ground ($R_{260m/ground}$) varied dramatically throughout the entire study with the daily average ranging from 0.42 to 1.10. In contrast, the total volume concentrations showed much better correlations between ground level and 260 m ($r^2 = 0.89$) and the average ratio was close to

one. Such differences were mainly caused by the different contributions of different mode particles to the number and volume concentrations.

The correlations of particle number and volume concentrations between ground level and 260 m varied substantially for different mode particles. As shown in Fig. 2a, the small Aitken mode particles were correlated between the two heights ($r^2 = 0.66$), indicating their common sources that are related to new particle formation. However, the average number concentration

at 260 m (1382 $\pm 1281$ cm$^{-3}$) was only approximately 40% of that at the ground level (3379 $\pm 2232$ cm$^{-3}$), and the daily average ratio of 260 m to ground level for $N_{15-40}$ varied from 0.91 to 0.51. These results illustrated additional sources for small Aitken particles at ground level. Indeed, pronounced peaks for $N_{15-40}$ were often observed at evening time, likely indicating the influences of local emissions, e.g., cooking and traffic emissions. The large Aitken mode particles showed the worst correlation between ground level and 260 m ($r^2 = 0.40$, slope = 0.70) although the average number concentrations were comparable (4188

vs. 3233 cm$^{-3}$). These results suggested the sources of large Aitken mode particles were quite different between ground level and 260 m. For example, the diurnal cycle of large Aitken mode particles at ground level was remarkably similar to that of COA (Zhao et al., 2016), likely indicating a large source contribution from cooking emission. Compared with Aitken particles, the number and volume concentrations of Accumulation mode particles were well correlated between the two heights ($r^2 = 0.85$ and 0.91, respectively). While the average number concentration at 260 m was 11% higher than that at ground level, the volume

concentration was close. Moreover, the temporal variations of accumulation mode particles tracked well with those of



secondary inorganic species that were mainly formed over a regional scale. Our results indicate that accumulation mode particles were likely dominantly from regional transport and relatively homogeneously distributed across different heights. The different vertical ratios between number and volume concentrations suggest that the particle size distributions were slightly different between ground level and 260 m.

The regional emission control showed a significant impact on particle number size distributions. As shown in Figs. 1a and 1b, the GMD of number size distributions peaked at 57 nm at 260 m and 43 nm at ground level, respectively during the control period, and the average size distribution showed three similar modes between the two heights. In contrast, the size distributions had substantial changes after the control period which were characterized by much broader distributions and clear shift from smaller to larger particles at both ground level and 260 m. For example, the GMD of particle number distributions was 106 nm

at 260 m which was much larger than that during the control period, and consistently the largest mode dominated the total number of particles, on average accounting for 68%. Figure 4 shows a comparison of average number and volume concentration between control and non-control periods for three mode particles. While the average total number concentrations during control period were lower than those during non-control periods (6139 vs. 8116 $cm^{-3}$ at 260 m, and 8708 vs. 10824 $cm^{-3}$ at ground level), the small and large Aitken mode particles were comparable between control and non-control periods. As a

result, the decreases in total number concentrations were mainly caused by the changes in accumulation mode particles which were decreased by 53% at 260 m and 52% at ground level during the control period. Our results illustrate that regional emission control has a large impact on accumulation mode particles while the influences on Aitken mode particles were small. One of the major reasons is that emission controls decrease the gas precursors (e.g., $SO_2$ and $NO_x$) and $PM_{2.5}$ mass concentrations substantially, and hence suppress the growth of particles to larger sizes. This is also consistent with the large decreases of

condensation sink (CS) by 48% at 260 m and 45% at ground level during the control period (Figs. 4a and 4b). In contrast, the frequency of new particle formation events showed an increase due to the lower PM loadings and the more clean days during the control period, leading to relatively comparable number concentrations of small particles with those after the control period. To better evaluate the impacts of regional emission controls, cluster analysis with hourly back trajectories were performed on the entire dataset with an exclusion of precipitation days. As shown in Fig. S4, accumulation mode particles during the control

period   showed the largest reductions for cluster 1 and 2 (39 and 42%, respectively) while the large Aitken particles had small changes and the small Aitken ones even showed a large increase (43%) for cluster 1. These results further support our conclusion above.

We also compared the particle number size distributions between polluted ($PM_{2.5} > 75$ μg $m^{-3}$) and clean days ($PM_{2.5} < 75$ μg $m^{-3}$) after the control period. As shown in Fig. S5, the average size distribution in polluted days at ground level showed a

clear three mode distribution, peaking at 36, 96 and 244 nm, respectively. The GMD of three modes was ubiquitously larger





than those (23, 41 and 106 nm) observed during clean days. While the average total number concentration was increased from 10258 (±4676) cm$^{-3}$ during clean periods to 12156 (±4406) cm$^{-3}$ in polluted days (Fig. 4), we observed comparable concentrations for small and large Aitken mode particles. Therefore, the increase in total number concentration was mainly caused by the accumulation mode particles which were increased by 90% during the polluted days. These results illustrate the

different roles of different mode particles between clean and polluted days. Similarly, the average particle number distribution showed a clear shift from smaller size during clean periods to larger size in polluted days at 260 m, and the total number concentration was increased by 53% from 7006 (±4416) to 10748 (±3615) cm$^{-3}$. Again, the increase in total number concentration was mainly due to the increase in accumulation mode particles by 135%. Compared with the number concentrations, the increases in volume concentrations for accumulation mode particles were more significant in polluted days,

which on average were 174% and 212% at ground level and 260 m, respectively. Indeed, the accumulation mode particles accounted for 97% of total volume concentrations at both ground level and 260 m, elucidating their major roles in PM pollution. The average number ratios between 260 m and ground level increased as a function of particle sizes during both clean and polluted days. For example, the ratios increased from 0.4 to 0.9 for small and large Aitken mode particles, and to 1.2 for accumulation mode particles in polluted days. These results are consistent with our previous conclusion that smaller particles

showed stronger vertical gradients than larger particles. We also observed ubiquitously higher $R_{260m/ground}$ in polluted days than clean periods, indicating larger vertical gradients in both number and volume concentration during polluted periods.

**3.2 Diurnal Variations**

The average diurnal variations of particle number size distribution at ground level and 260 m for the entire study are shown in Fig. 5. It is clear that particle number size distributions show pronounced diurnal cycles which were characterized by

the lowest values in the early morning and subsequent particle growth until midnight. During the growth period, the GMD increased from 29 to 57 nm in 14 h at ground level, while it increased from 41 to 88 nm in 12 h at 260 m. After that, the GMD remained at relatively constant levels at both ground level and 260 m, which are 70 and 100 nm, respectively. The ubiquitously lower GMD and lower growth rates at ground level were likely due to the influences of local emissions that contain a large amount of small particles. Noted that the changes in GMD were significant at ground level after the control period, especially in

polluted days (Fig. S6), indicating that the diurnal evolution of particle number size distributions at ground level is subject to multiple influences. In contrast, the changes at 260 m were much smaller with a relatively consistent mode peaking at ~100 nm, indicating a more constant particle source at higher altitudes.

The particle number ratios between 260 m and ground level depend strongly on particle size. As shown in Fig. 5c, $R_{260m/ground}$ increases rapidly between 15 – 100 nm as the increase of particle size but typically less than one. This is consistent





with our previous conclusion that small particles are more abundant at ground level due to the influences of local emissions. $R_{260m/ground}$ increases continuously and reached a maximum at $D_p$ = ~150 nm. One explanation is the faster condensational and coagulational growth of small particles at 260 m than ground site. Another explanation is the enhanced regional transport of 100 – 200 nm particles at high altitude. This is consistent with the fact that much higher $R_{260m/ground}$ was observed during polluted periods than clean periods. $R_{260m/ground}$ decreased to less than 1 at $D_p$ > 250 nm, likely due to the deposition of large particles.

Our results show that the vertical differences in particle number concentrations varied significantly as a function of size, which has important implications that the health and climate effects of aerosol particles at different heights could be substantially different.

The diurnal cycles of particle number and volume concentrations at 260 m and ground level, as well as $R_{260m/ground}$ during different periods are illustrated in Figs. 6 and S7. Pronounced diurnal cycles with two clear peaks at noon and evening time

were observed at both ground level and 260 m. Further analysis highlight that these two peaks were driven by small and large Aitken mode particles, respectively (Figs. 6b and 6c), likely representing two dominant sources of new particle formation and cooking emissions, respectively. In comparison, the diurnal cycles of accumulation mode particles were relatively flat indicating the sources were mostly regional. Figure 6 shows that the total particle number concentration during the control period was consistently lower than that after the control period, particularly during the time period of 0:00 – 8:00. Such

decreases were mainly caused by accumulation mode particles which were reduced by 32 – 67% at ground level and 23 – 69% at 260m, respectively, throughout the day. In contrast, the diurnal cycle of small Aitken mode particles was substantially different, which is characterized by a prominent peak between 10:00 – 14:00 associated with new particle events, and a smaller second peak at nighttime due to the influences of local emissions. The particle number concentration of the NPE peak during the control period was even higher than that after the control period, while the difference at nighttime was much smaller. These

results suggest that regional emission controls could increase the number of small particles while decrease accumulation mode particles significantly. One explanation is that the growth of small particles was suppressed due to the lower concentrations of precursors and PM loadings. The diurnal cycles of $R_{260m/ground}$ for different sizes were overall similar during and after the control period, which are all characterized by clear daytime increases due to enhanced vertical mixing, and subsequent decreases at nighttime due to more influences of local source emissions on ground site.

We also compared the diurnal cycles of particle number concentrations between clean and polluted periods. Again, very different diurnal profiles were observed for particles in different size ranges. While small Aitken mode particles at 260 m showed clear daytime increases during both clean and polluted periods, those at ground site however varied more dramatically due to the influences of multiple sources. Similarly, the total number of small Aitken mode particles was slightly lower during polluted periods compared to clean periods. In contrast, the diurnal cycles of large Aitken mode particles were quite different





between ground level and 260 m. While a pronounced nighttime peak due to cooking influences was observed at ground level,

more diurnal peaks that were associated with different sources and processes were observed at 260 m. The largest difference

between clean and polluted periods was observed during 0:00 – 8:00 at 260 m, while it was much smaller at ground level. Such

differences clearly indicate very different vertical gradients between clean and polluted periods for large Aitken mode particles.

Compared to Aitken mode particles, the number concentration of accumulation mode particles during polluted periods was

more than a factor of ~2 – 3 of those during clean periods. These results suggest that the major difference of particle number

characteristics between clean and polluted periods is accumulation mode particles. In fact, the CS during polluted periods was

nearly twice that of during clean periods (Fig. 4), which facilitated the growth of particles.

**3.3 Chemistry of particle growth**

Particle growth events (NPE) were frequently observed during the entire study at both ground level and 260 m. As shown

in Fig. 7, the growth process of particles at ground level started from approximately 9:00 until mid-night with the GMD

increasing from ~22 nm to ~60 nm. This result was consistent with those previously observed at urban and rural sites in Beijing

(Wang et al., 2013a). Similarly, the growth of particles started from ~28 nm at 9:00 to ~63 nm at mid-night at 260 m. The

growth of particles was closely related to the diurnal cycle of CS, which showed a continuous increase from early morning to

mid-night. Also, aerosol composition had significant changes during the growth periods. As indicated in Figs. 7b and 7d, the

contribution of organics first showed an increase during the early growth period between 8:00 – 12:00, while those of other

chemical species remained small changes. After 12:00, both organics and sulfate showed increased contributions until 17:00.

Although the increases in organics and sulfate were partly due to the decreases in nitrate and chloride because of the

evaporative loss in the afternoon, our results likely indicate that organics played an important role in the early stage of particle

growth, while both organics and sulfate are important in the subsequent growth.

We further calculated the particle growth rates (GR) for each growth events that lasted more than 3 hours (Fig. 8). The

particle GR varied from 1.4 nm h$^{-1}$ to 7.5 nm h$^{-1}$ at 260 m and from 1.5 nm h$^{-1}$ to 6.1 nm h$^{-1}$ at ground level, which generally

falls within the range that was reported previously in various environments (Kulmala et al., 2004), e.g., Beijing (Wu et al.,

2007;Yue et al., 2010;Zhang et al., 2011), Shangdianzi (Shen et al., 2011), Egbert (Pierce et al., 2014), Marseille (Kulmala et al.,

2005), and New Delhi (Sarangi et al., 2015). Particle growth rates strongly depend on temperature and the availability of

condensable vapors. Indeed, the particle GR in the study was generally correlated well with CS at both ground level and 260 m

during periods with low sulfate concentrations (Figs. 8d and 8e). The average particle GR was 3.6 nm h$^{-1}$ at 260 m, which is

slightly higher than 3.3 nm h$^{-1}$ at ground level, which is likely due to the lower temperature at high altitude. It is interesting to

note that GR was correlated with the change of organic concentration (ΔOrg) at 260 m, and also correlated well with that during





periods with low sulfate concentrations (e.g., < 3μg m$^{-3}$) at ground, likely indicating a dominant role of organics in the growth.

As shown in Fig. 8a, high sulfate concentrations were generally observed during polluted periods with high PM loadings, and

corresponding, relatively higher GR was related to higher sulfate concentration. Our results here suggest that the particle

growth mechanisms could be different between clean periods with dominance of organics and polluted periods with

significantly enhanced sulfate.

### 3.4 Source apportionment

PMF analysis of size-resolved particle number concentrations was able to identify four and five factors at 260 m and

ground level, respectively (Fig. 9). The five factor solution at 260 m yielded a split factor that cannot be physically interpreted.

The average number size distributions of factor 1 showed GMDs peaking at 20 and 27 nm at ground level and 260 m,

respectively, and the temporal variations were characterized by frequent sharp peaks in most days (Fig. 9c). It is clear that this

factor was associated with new particle events. This is further supported by the pronounced diurnal cycles showing rapid

increases between 8:00 – 12:00, and a dominant source region to the west (Fig. S9a), where clean air masses were prevalent.

However, we also noticed the differences in diurnal cycles between ground level and 260 m. For example, the diurnal cycle of

factor 1 at the ground site showed two peaks during morning and evening traffic hours, likely indicating the influence of traffic

emissions. In fact, the time series correlation between the two heights was weak (r$^2$ = 0.17), confirming that the sources of

factor 1 are not the same. The average particle number concentration of factor 1 was 816 and 1067 cm$^{-3}$ at ground site and 260

m during the control period, which was even 31% and 38% higher than those after the control period. One explanation is due to

the increase of CS after the control period that facilitated the condensation and coagulation of small particles. Our results also

suggest that regional emission controls could increase the number of nucleation mode particles because the reduction of

precursors and PM loadings.

Factor 2 presented a size distribution peaking at ~32 nm and a distinct diurnal cycle with two comparable and pronounced

peaks at noon and evening time. The diurnal cycle of factor 2 resembled that of cooking organic aerosol that was widely

reported in Beijing (Huang et al., 2010;Sun et al., 2013;Zhang et al., 2016;Elser et al., 2016;Xu et al., 2015). On average, this

factor accounted for 25% of the total particle number concentration, and had only a small difference (2%) between control and

non-control period. Likely, this factor was dominantly contributed by cooking emissions although particle growth can explain

partly the high concentrations during the late afternoon. Factor 3 at ground level showed a similar diurnal cycle as factor 2, yet

the evening peak was much higher than noon peak. Such a diurnal profile was remarkably similar to that of COA that was

resolved from PMF analysis of OA during the same study period (Zhao et al., 2016). Also, the particle number size distribution

of factor 3 was similar to that from cooking activities (Buonanno et al., 2011). These results supported that factor 3 was mainly



from cooking emissions. Similar to factor 2, there was only a small change (3%) during and after the control period, consistent

with the fact that no control measures were implemented near our sampling site during the control period. Compared to the

ground site, factor 3 at 260 m also showed two pronounced peaks in the diurnal profile. However, the nighttime peak was much

smaller than that at ground level. This can be explained by the significantly enhanced cooking emissions at nighttime at ground

level, yet the vertical mixing to high altitude is limited due to the average number concentration at ground level was 3375 cm$^{-3}$,

which was 64% higher than that at 260 m, indicating stronger influences of local cooking emissions on particle numbers at

lower altitudes. This factor was moderately correlated between ground level and 260 m ($r^2$ = 0.37), indicating that cooking

sources could be also different at different altitudes, for example, more contributions from regional cooking emissions at higher

altitudes. In addition, factor 3 at 260 m was better correlated with the sum of factor 2 and factor 3 at ground level ($r^2$ = 0.40, Fig.

S8), further supporting that these three factors have similar sources. Another evidence is that factor 2 and 3 have the smallest

influences from regional emission control among all factors.

Factors 4 and 5 showed quite different temporal variations, but were generally characterized by high concentrations during

polluted periods. As shown in Fig. 9, the time series of factor 4 was highly correlated between ground level and 260 m ($r^2$ =

0.74) although the peak diameter in size distributions was slightly different (114 and 98 nm, respectively). These results suggest

a similar source of factor 4 at different altitudes. The diurnal cycle of factor 4 was also similar at the two different heights which

both showed a small noon peak and high concentrations at night. Such a diurnal cycle was much similar to that of less oxidized

SOA observed during the same study (Zhao et al., 2016). Therefore, we inferred that factor 4 is a secondary factor that was

associated with photochemical processing and semi-volatile species. Compared to factor 4, factor 5 showed the best correlation

between the two heights ($r^2$ = 0.91), and the time series and diurnal cycles were remarkably similar to those of highly oxidized

SOA and sulfate (Zhao et al., 2016), indicating that factor 5 is an aged secondary factor and was mainly formed over a regional

scale. Consistently, the bivariate polar plot of factor 5 showed a dominant source region to the south, supporting a major

influence of regional transport from the south. Regional emission controls showed large yet different impacts on factor 4 and

factor 5. While the average number concentrations of factor 4 showed decreases by 49% and 37% at ground level and 260 m,

respectively during the control period, those of factor 5 had the most reductions by 65% and 74%, respectively. These results

are consistent with our previous conclusions that regional emission controls have the most impacts on highly aged secondary

aerosols (Sun et al., 2016;Zhao et al., 2016).

Overall, the five factors represent the major sources of particle numbers in the megacity of Beijing, which are associated

with new particle events, local primary emissions (e.g., cooking and traffic emissions), and secondary formation with different

aging process. The contribution of secondary sources was dominant at 260 m throughout the day by varying from ~50% to 80%

(Fig. 10b), and the average contribution (60%) was also higher than that (34%) at ground site. In contrast, the cooking source



was the largest contributor to the total particle numbers, on average accounting for 33%. Therefore, our results not only illustrated the similarities and differences of particle number concentrations and sources at different altitudes in megacity, but

also demonstrated the different responses of sources factors to regional emission controls.

**4 Conclusions**

We conducted the first simultaneous real-time measurements of particle number size distribution along with aerosol particle composition at ground level and 260 m on a meteorological tower in urban Beijing from 22 August to 30 September, 2015. Our results showed that the number size distributions had significant differences between the two heights although the

particle volume and $PM_1$ mass concentrations were overall similar. The average number concentration (15 – 400 nm) was 7473 ($\pm$4324) $cm^{-3}$ at 260 m, which is 26% lower than that at ground level (10134 ($\pm$4680) $cm^{-3}$). The number concentrations of Accumulation particles (100 – 400 nm) at 260 m was highly correlated with those at ground level ($r^2$ = 0.85), indicating their similar sources. However, the correlations were much weaker for Aitken mode particles suggesting that they have more different sources at different altitudes. A more detailed analysis suggests that the vertical differences in particle number

concentrations varied as a function of sizes. While particles in the size range of 100 – 200 nm showed higher concentrations at 260 m, those of smaller particles were more dominant at ground level. These results might indicate the different contributions of local emissions and regional transport to particle numbers at different altitudes. We also observed an increase of the ratio of 260 m to ground for all particles in different size ranges during daytime, highlighting the impacts of vertical mixing.

Particle growth events were occasionally observed in this study. The average particle growth rate was 3.6 nm $h^{-1}$ at 260 m

and 3.2 nm $h^{-1}$ at ground level, respectively. By comparing with aerosol composition changes during the growth period, we found that organics appeared to play a more important role than sulfate during the early stage of the growth (9:00 – 12:00), while organics and sulfate are both important after that. The sources of particle numbers were characterized by PMF, and our results illustrated three common sources at different altitudes, i.e., new particle formation and growth, local secondary formation, and regional transport. We also observed much higher primary emissions from cooking sources at ground level than

260 m, highlighting the importance of local sources emissions in characterization of NPF and growth events at ground level. In addition, we found that regional emission controls exerted a large impact in reducing accumulation mode particles, for example, by 65-74% for the regional factor, while had minor impacts on small Aitken mode particles mainly due to the enhanced NPF events and the limited controls on local source emissions. These results are overall consistent with the conclusions from our previous studies during the Asia-Pacific Economic Cooperation (Sun et al., 2016;Xu et al., 2015;Chen et al., 2015).






*Acknowledgements.* This work was supported by the National Key Project of Basic Research (2014CB447900, 2013CB955801), the National Natural Science Foundation of China (41575120), the Special Fund for Environmental Protection Research in the Public Interest (201409001), and the Strategic Priority Research Program (B) of the Chinese Academy of Sciences (XDB05020501).

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





Table 1. The geometric mean diameter (GMD) of average particle number size distribution for different periods at ground level and 260 m. Also shown are GMDs of three modes from log-normal fitting.

| GMD | Entire Study | | Three Modes | | | | | |
| | 260 m | Ground | 260 m | | | Ground | | |
|---|---|---|---|---|---|---|---|---|
| Entire Study | 88 | 45 | 27 | 44 | 116 | 24 | 41 | 111 |
| Control Period | 57 | 43 | 27 | 48 | 104 | 24 | 46 | 150 |
| non-Control Period | 106 | 47 | 27 | 43 | 119 | 23 | 40 | 102 |
| Clean | 79 | 47 | 27 | 45 | 112 | 23 | 41 | 106 |
| Polluted | 131 | 47 | 52 | 113 | 188 | 36 | 96 | 244 |

Table 2. Summary of average number concentration of five factors for the entire study, control period (CP), non-control period (NCP), and also the change percentages (= (CP-NCP)/NCP×100).

| | F1 | | F2 | | F3 | | F4 | | F5 | |
| | 260m | Ground | 260m | Ground | 260m | Ground | 260m | Ground | 260m | Ground |
|---|---|---|---|---|---|---|---|---|---|---|
| Entire study (cm$^{-3}$) | 867 | 695 | - | 2567 | 2066 | 3376 | 2859 | 2662 | 1412 | 801 |
| Control period (cm$^{-3}$) | 1067 | 816 | - | 2586 | 2271 | 3314 | 2049 | 1619 | 489 | 357 |
| Non-control period (cm$^{-3}$) | 771 | 621 | - | 2526 | 1967 | 3413 | 3249 | 3162 | 1856 | 1023 |
| (CP-NCP)/NCP (%) | 38% | 31% | | 2% | 15% | -3% | -37% | -49% | -74% | -65% |





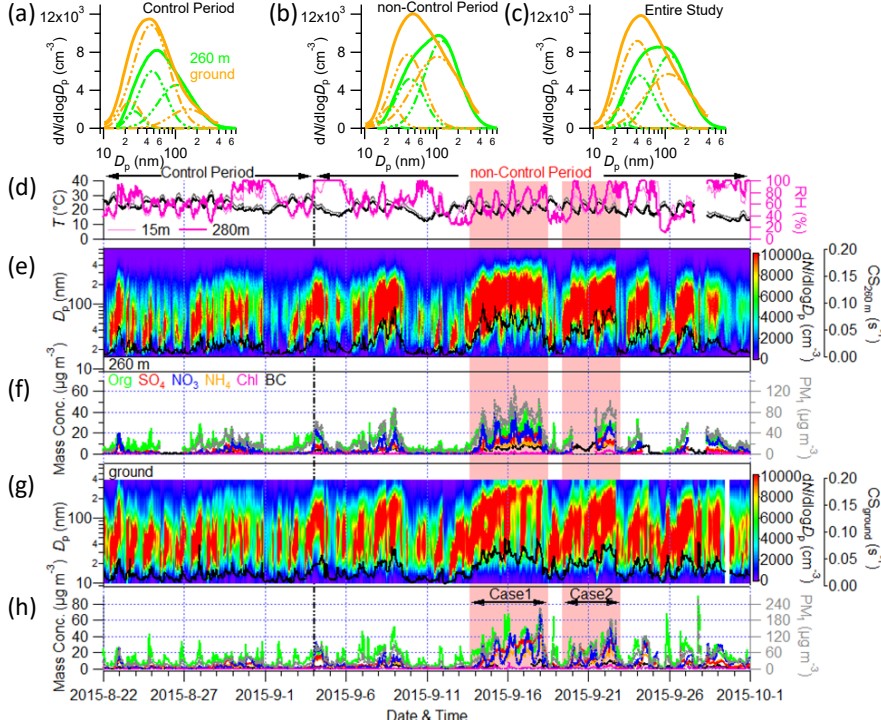

Figure 1. Average particle number size distributions during (a) control period, (b) non-control period, and (c) the entire study at ground level

(orange lines) and 260 m (green lines). (d) shows the time series of meteorological parameters of relative humidity (RH) and temperature ($T$).

(e) and (g) are the particle number size distributions and condensation sink (CS) at 260 m and ground level, respectively. (f) and (h) are the

time series of mass concentrations of $PM_1$ species at 260 m and ground level, respectively.





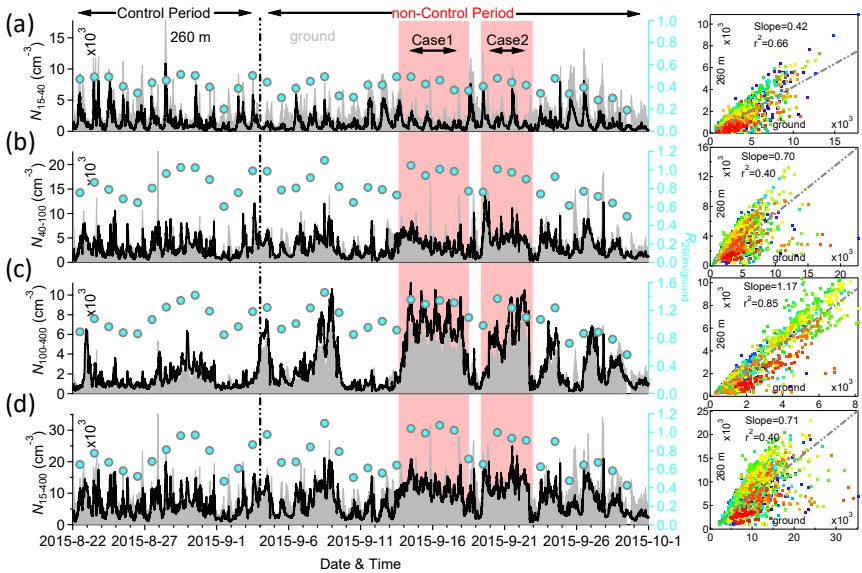


Figure 2. Comparisons of particle number concentrations between ground level and 260 m for different size ranges, i.e., (a) small Aitken mode (15 – 40 nm), (b) large Aitken mode (40 – 100 nm), (c) Accumulation mode (100 – 400 nm), and (d) the total number of particles (15 – 400 nm). Right figure show the scatter plots of the comparisons.

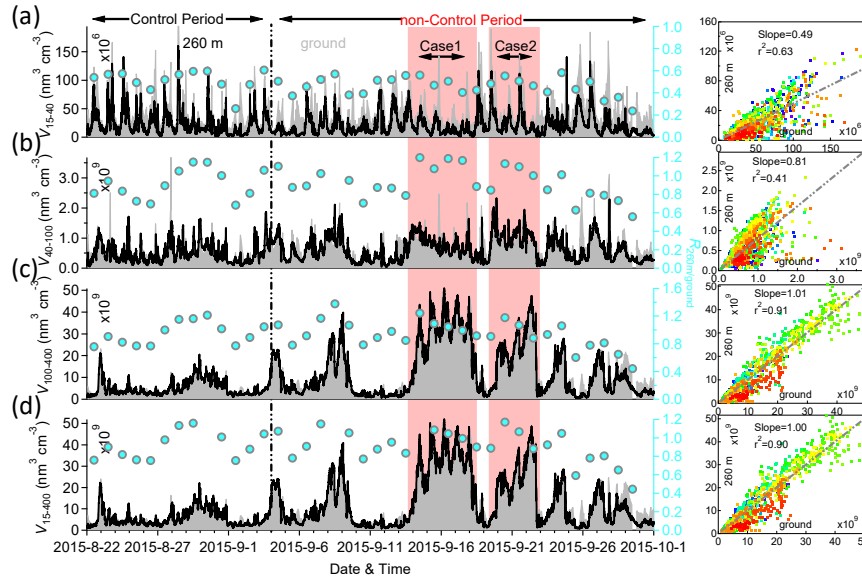


Figure 3. Comparisons of particle volume concentrations between ground level and 260 m for different size ranges, i.e., (a) small Aitken mode (15 – 40 nm), (b) large Aitken mode (40 – 100 nm), (c) Accumulation mode (100 – 400 nm), and (d) the total number of particles (15 – 400 nm). Right figure show the scatter plots of the comparisons.





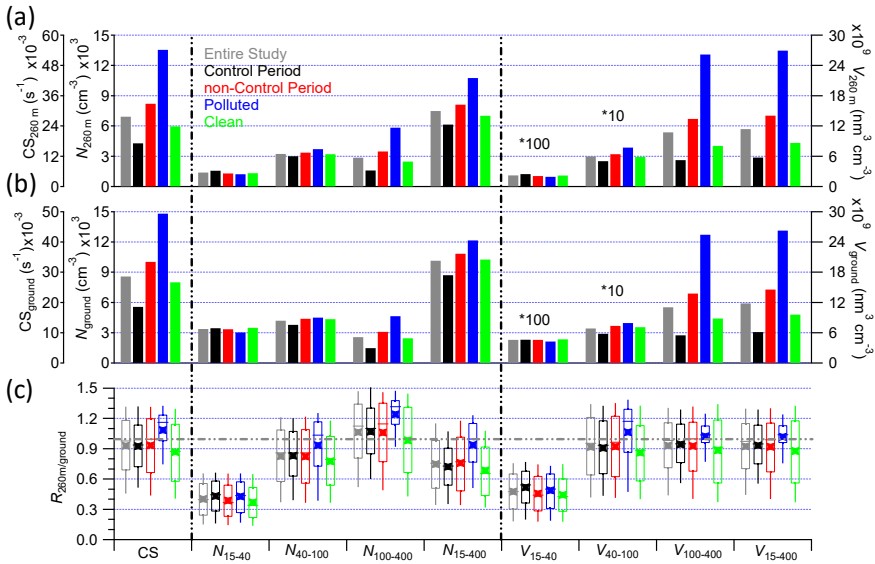

Figure 4. Average number and volume concentrations, and CS at (a) 260 m and (b) ground level for the entire study and four different periods.

(c) shows the box plots of the ratios of 260 m to ground level. The volume concentrations of small and large Aitken modes are enhanced by a factor of 100 and 10, respectively for clarity.

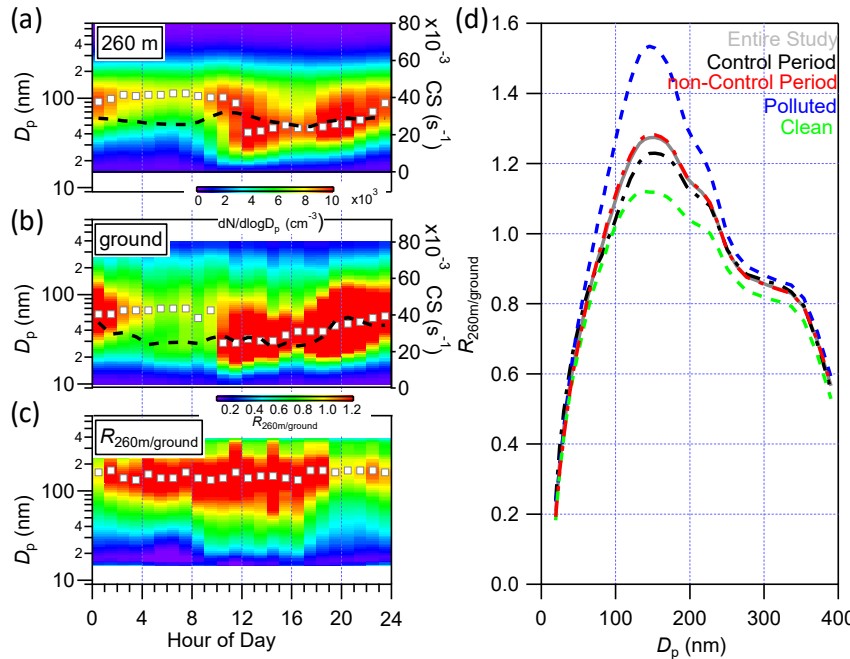

Figure 5. Average diurnal variations of particle number size distributions at (a) 260 m and (b) ground level, and (c) the ratios of 260 m to ground level for the entire study. (d) shows the ratios of particle number concentrations at 260 m to those at ground level as a function of





particle sizes.

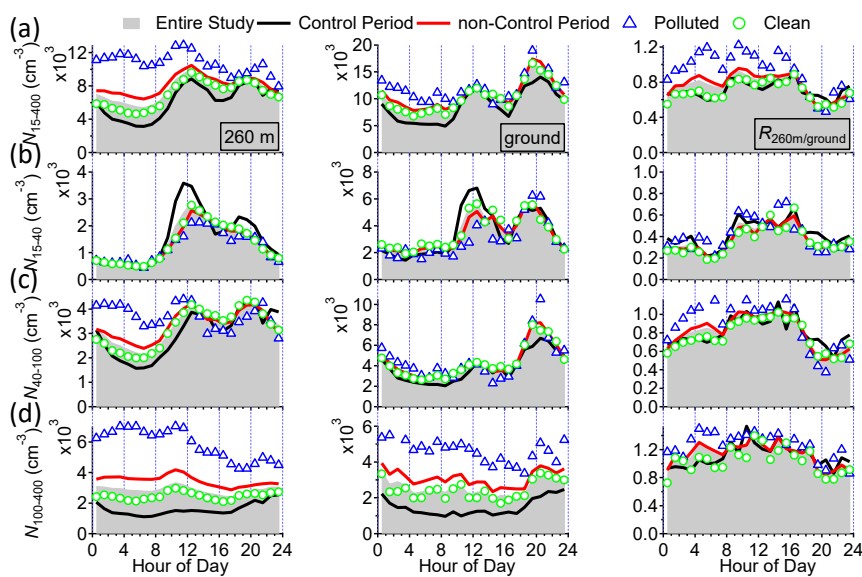

Figure 6. The diurnal cycles of particle number concentrations at 260 m and ground level, and the ratios of 260 m to ground for different size ranges, i.e., (a) 15 – 400 nm ($N_{15–400}$), (b) small Aitken mode ($N_{15–40}$), (c) large Aitken mode ($N_{40–100}$), and (d) Accumulation mode ($N_{100–400}$).

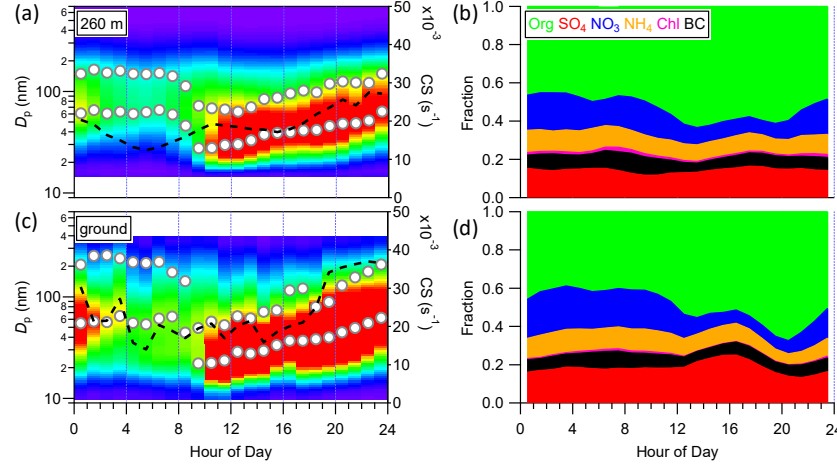


Figure 7. Average diurnal evolution of particle number size distributions and aerosol composition at (a,b) 260 m and (c,d) ground level for the new particle growth events. The dash lines in (a) and (c) are the diurnal cycles of CS.



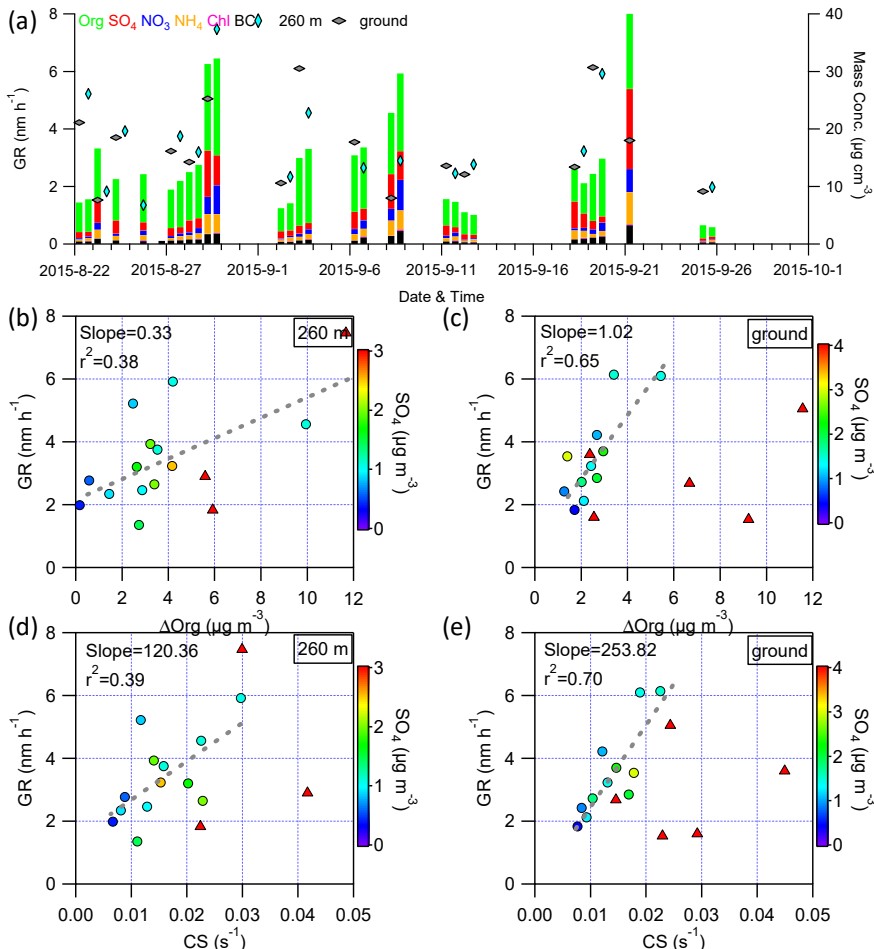

Figure 8. (a) Time series of particle growth rates and corresponding average chemical composition for selected particle growth events. (b) and
(c) show the correlation of particle growth rates with the changes in the concentration of organics (ΔOrg) at 260 m and ground level,
respectively. (d) and (e) show the correlation of particle growth rates with condensation sink at 260 m and ground level, respectively. The data
points in (b-e) are color coded by the mass concentration of sulfate ($SO_4$), and those with sulfate concentrations higher than 3 µg m$^{-3}$ (ground
level) and 2.5 µg m$^{-3}$ (260 m) are marked as triangle points.

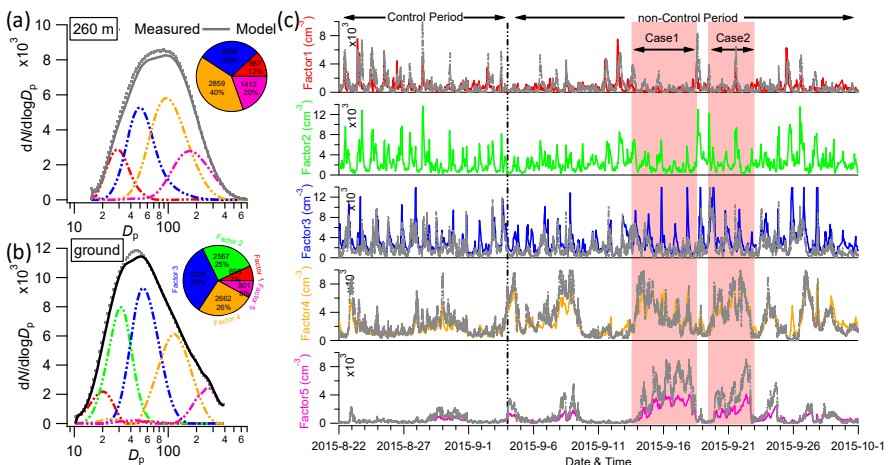

Figure 9. (a) and (b) Factor profiles of particle number size distributions at 260 m and ground level, respectively. (c) Comparisons of the time series of PMF factors at 260 m (gray dash lines) and ground level (color coded lines).

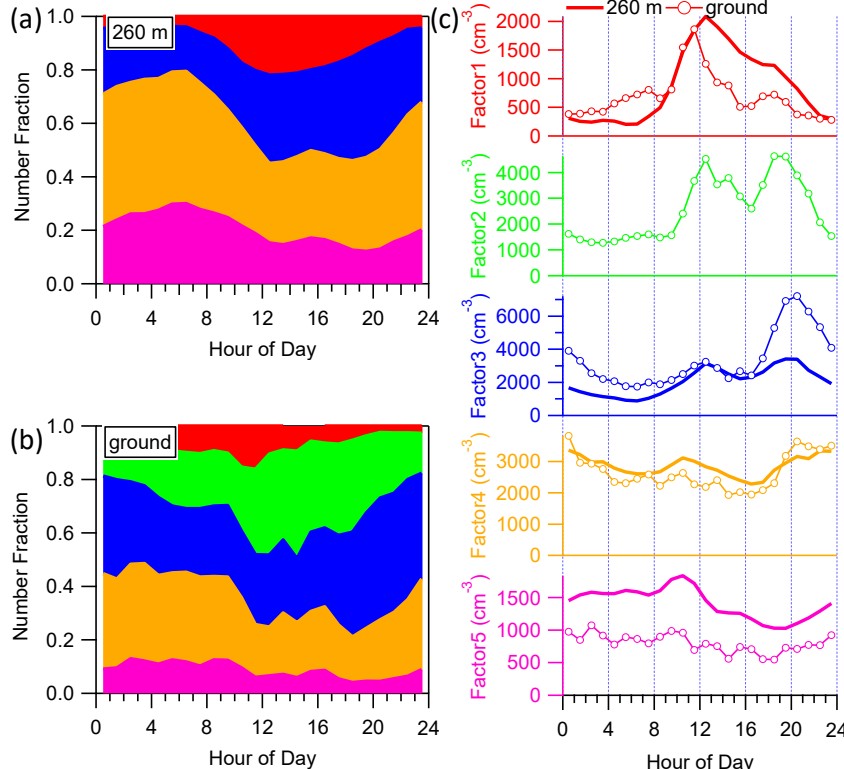

Figure10. Average diurnal variations of number fraction of PMF factors at (a) ground level and (b) 260 m. (c) shows a comparison of the average diurnal cycles of particle number concentrations for PMF factors at ground level and 260 m.