# Peer review of "Simultaneous measurements of particle number size distributions at ground level and 260 m on a meteorological tower in urban Beijing, China"

_Atmospheric Chemistry and Physics, 2016_

## Referee Comment (RC1) · Anonymous Referee #2 · 25 Jan 2017

This manuscript presents simultaneous measurements of particle number size distribution and particle chemical composition at a high level of 260 meter and at ground level. Information on comparison measurements in megacities such as Beijing can provide new insights into vertical distribution of particle formation and growth. In addition, comparison of measurements between control and non-control period will help to evaluate the effectiveness of the proposed emission control strategies. The paper in general is well written and should be publishable after some minor issues are address:

1. The authors conclude that investigation of new particle formation and growth events

at ground level in megacities needs to consider the influences of local cooking emissions. This is a very strong statement. Can the conclusion be generalized in megacities around the world or it is just constrained to some regions?

2. The conclusion of emission controls enhancing new particle formation is another strong statement that may need a little more elaboration, for example, more in-depth data analysis and showing more evidences.

3. The authors claimed that "One of the major reasons is that emission controls decrease the gas precursors (e.g., SO2 and NOx) and PM2.5 mass concentrations substantially and hence suppress the growth of particles to larger sizes." Is there something else that needs to be explored? For example, is it possible that accumulation mode particles are controlled from their direct emissions from industries or coal-fired power plants in addition to their contribution from secondary formation?

4. The authors used PMF to perform source apportionment and found two factors (factors 2&3) is likely associated with cooking emissions. The size distribution corresponding to factor 2 peaks at about 32 nm, which by and large falls within the small Aitken mode size range (15-40nm). This conclusion somehow is not consistent with the statements between L160-170 which attribute large Aitken mode particles (40-100 nm) to cooking emissions. In addition, while both factors 2 and 3 are attributed to local cooking emissions, what are the reasons that they are divided into two factors rather than combined into one?

---

## Referee Comment (RC2) · Anonymous Referee #3 · 25 Jan 2017

This is an interesting work and this referee has a few minor comments for authors considering. 1) Lines 48-51, "The first continuous measurements of aerosol number size distributions within the city area of Beijing indicated a high variability in number concentrations, and the variations were substantially different among dust storm, clean and polluted periods (Wehner et al., 50, 2004)." This is not fact, please double check and give a credit to a right one. 2) Lines 56-57 "organics were found to be the dominant species in new particle formation events during the Beijing Olympic Games (Zhang et al., 2011)". No direct measurements for chemicals in <50 nm atmospheric particles were available in China, how can Zhang find organics to be the dominant species in

new particle formation events? Argue? 3) Lines 63-64 "Therefore, measurements of size-resolved number concentrations at high altitude with less local influences" Why? Local stacks at height can also greatly increase particle number concentrations? If the sampling site is on the route for air plane landing or taking off, huge local emissions at height are also there. 4) Lines 84-92, SMPS suffers from a problem in accurately measuring particle size distribution in dynamic polluted air and is also unable to separate primaryy particles from grown new particles in the size range > 30 nm. The weakness should be considered and mentioned. 5) Lines 135-145, the referee has tested size dsitributions of particle number cocnentration and found that there was a dominant mode at ~20 nm. Of course, different cookings may not generate the same size dsitributions of particle number cocnentration. Please give more evidences for cooking source. 6) Lines 163-164 "Indeed, pronounced peaks for N 15-40 were often observed at evening time, likely indicating the influences of local emissions, e.g., cooking and traffic emissions." Yes, the two types of sources could be the cause. Vertical exchange of regional trasnported particles can also be a potentail cause. 7) Lines 175-195, changing size distributions of particle number concentrtion between two periods can also be due to the presence or absence of cloud-modification and should be considered. More clear days in control periods even strongly implied the possibilty. 8) Lines 220-221 "During the growth period, the GMD increased from 29 to 57 nm in 14 h at ground level, while it increased from 41 to 88 nm in 12 h at 260 m" It could be true, but hard to believe this. Please consider the weakness of SMPS measurments in dynamic urban atmospheres. 9) "our results likely indicate that organics played an important role in the early stage of particle growth, while both organics and sulfate are important in the subsequent growth." Without direct measurements for chemicals in nucleation mode particles, it is really hard to say this. The same comment is applicable for lines 290-293. 10) Section 3.4, please consider cloud-modication for particle number size distribution

---

## Referee Comment (RC3) · Anonymous Referee #1 · 27 Feb 2017

This manuscript reported the first simultaneous observation of particle size distribution at two different height (260m and ground level) in the megacity of Beijing during periods with or without emission control. The aerosol chemical composition was also reported and connected to particle growth. This work did provide useful information to understand the particle nucleation and growth in the PBL. The manuscript is overall well written and fits the scope of ACP. I recommend it can be published on ACP after some miner revision.

1) New particle formation and growth events are generally abbreviated as NPF, not
NPE.

2) ACSM can only measure the chemical composition of particles larger than several tens of nanometer. The authors need to be very careful to use ACSM measurement to explain the initial growth of newly formed particles.

3) Line 167, what is COA?

4) The reduction of PM2.5 would in general promote the new particle formation and growth due to the decreasing of condensation sink. In this MS, e.g. 187-189 and 250-252, the author attributed lower growth rates to lower PM loading. This is an unreasonable explanation.

5) Line 266-267: Similar as last comment, higher CS should suppress the particle growth.

6) Line 286-287: it's better to compare the GR event to event, but not the average value.

7) Figure 2 and Figure 3: the color bars are missing.

8) It would be good to change Figure 4 to a table.

9) Figure 5: the data with higher time resolution, e.g. 10 min is recommended for figure a, b and c.

---

## Author Comment (AC1) · 10 Apr 2017

**Response to the Reviewers' comments**

We are thankful to the three reviewers for their constructive comments that help improve the manuscript significantly. Following the reviewer's suggestions, we have revised the manuscript accordingly. Listed below are our point-by-point responses in blue to each reviewer's comments

**Response to Reviewer #1**

This manuscript reported the first simultaneous observation of particle size distribution at two different height (260 m and ground level) in the megacity of Beijing during periods with or without emission control. The aerosol chemical composition was also reported and connected to particle growth. This work did provide useful information to understand the particle nucleation and growth in the PBL. The manuscript is overall well written and fits the scope of ACP. I recommend it can be published on ACP after some miner revision.

We thank the reviewer's positive comments.

1) New particle formation and growth events are generally abbreviated as NPF, not NPE.

We thank the reviewer's comment. Because AMS and ACSM can only detect particles with aerodynamic diameter larger than 30 nm, and the SMPS measurements in this study are above 15 nm in mobility diameter, our study mainly focus on characterization of the later stage of the particle growth. We then used NPE to represent the abbreviation of new particle growth events after new particle formation.

2) ACSM can only measure the chemical composition of particles larger than several tens of nanometer. The authors need to be very careful to use ACSM measurement to explain the initial growth of newly formed particles.

We agree with the reviewer that interpretation of the early particle growth with the ACSM data should be cautious because 1) ACSM can only detect particles with aerodynamic diameter larger than 30 nm, 2) relatively lower sensitivity (higher detection limits) compared with the research-grade AMS.

3) Line 167, what is COA?

COA is cooking organic aerosol, which was spelled out in the revised manuscript.

4) The reduction of PM2.5 would in general promote the new particle formation and growth due to the decreasing of condensation sink. In this MS, e.g. 187-189 and 250-252, the author attributed lower growth rates to lower PM loading. This is an unreasonable explanation.

Right, the reduction of $PM_{2.5}$ would in general promote the new particle formation and early particle growth, but would also decrease the later particle growth because of the less pre-existing particles. This is also consistent with our observations of the decreases in condensation sink.

5) Line 266-267: Similar as last comment, higher CS should suppress the particle growth.

We thank the reviewer's comment. Because the limited particle sizes measured in this study, we mainly focus on characterization of particle growth after 20 nm. While higher CS can suppress new particle formation, it can also enhance the condensation on pre-existing particles and increase particle growth.

6) Line 286-287: it's better to compare the GR event to event, but not the average value.

We thank the reviewer's comment. We checked the GR of each event before presenting the average values. Except the days of 9/3, 9/6, 9/11, 9/19, the GR of newly formed particles at 260 m was ubiquitously higher than that at ground level. Table R1 shows the condensation sink, temperature

and relative humidity for new particle growth event. The CS at 260 m with higher relative humidity and lower temperature was always higher than that at ground level.

Table R1. The condensation sink (CS), temperature (T), relative humidity (RH) in particle growth events at two heights.

| | CS_ground $(S^{-1})$ | CS_260 $(S^{-1})$ | T_ground (°C) | T_260 (°C) | RH_ground (%) | RH_260 (%) |
|---|---|---|---|---|---|---|
| 8/22 | 0.012 | 0.012 | 30.3 | 27.1 | 39.0 | 43.8 |
| 8/23 | 0.023 | 0.022 | 26.0 | 24.1 | 61.5 | 67.0 |
| 8/24 | 0.015 | 0.014 | 28.6 | 25.8 | 44.7 | 49.8 |
| 8/25 | | 0.011 | | 26.0 | | 47.8 |
| 8/26 | | 0.012 | | 26.6 | | 46.4 |
| 8/27 | 0.013 | 0.016 | 29.6 | 26.7 | 40.4 | 45.1 |
| 8/28 | 0.017 | 0.020 | 30.8 | 27.7 | 36.2 | 41.1 |
| 8/29 | 0.024 | 0.030 | 29.1 | 25.4 | 46.0 | 54.8 |
| 9/2 | 0.009 | 0.008 | 29.6 | 26.7 | 39.9 | 44.5 |
| 9/3 | 0.019 | 0.023 | 30.6 | 27.0 | 39.2 | 46.7 |
| 9/6 | 0.018 | 0.023 | 26.3 | 23.4 | 42.2 | 48.7 |
| 9/8 | 0.029 | 0.042 | 26.7 | 23.8 | 37.2 | 41.9 |
| 9/11 | 0.010 | 0.013 | 21.7 | 19.0 | 46.1 | 52.3 |
| 9/12 | 0.008 | 0.009 | 23.5 | 21.1 | 26.7 | 27.5 |
| 9/18 | 0.015 | 0.015 | 29.2 | 26.4 | 30.7 | 33.9 |
| 9/19 | 0.023 | 0.030 | 27.5 | 24.7 | 29.4 | 32.6 |
| 9/21 | 0.045 | | 27.6 | | 40.0 | |
| 9/25 | 0.008 | 0.007 | 25.2 | 22.3 | 14.5 | 17.6 |

7) Figure 2 and Figure 3: the color bars are missing.

Thank the reviewer's carefulness. The color bars were added in Figures 2 and 3 in the revised manuscript.

8) It would be good to change Figure 4 to a table.

We thank the reviewer's comment. We added a Table in supplementary following the reviewer's suggestion while keeping this figure in the manuscript for easy reading,.

Table R2. Average CS, number and volume concentrations at 260 m and ground level for the entire study and four periods.

| | | CS $(s^{-1})$ | $N_{15-40}$ $(cm^{-3})$ | $N_{40-100}$ $(cm^{-3})$ | $N_{100-400}$ $(cm^{-3})$ | $N_{15-400}$ $(cm^{-3})$ | $V_{15-40}$ $(nm^3\ cm^{-3})$ | $V_{40-100}$ $(nm^3\ cm^{-3})$ | $V_{100-400}$ $(nm^3\ cm^{-3})$ | $V_{15-400}$ $(nm^3\ cm^{-3})$ |
|---|---|---|---|---|---|---|---|---|---|---|
| Entire Study | 260 m | 0.028 | 1382 | 3233 | 2858 | 7473 | 2.21E+07 | 5.96E+08 | 1.07E+10 | 1.14E+10 |
| | Ground | 0.029 | 3379 | 4188 | 2567 | 10134 | 4.58E+07 | 6.84E+08 | 1.11E+10 | 1.18E+10 |
| | $R_{260m/ground}$ | 0.93 | 0.40 | 0.83 | 1.06 | 0.75 | 0.48 | 0.92 | 0.93 | 0.92 |
| Control Period | 260 m | 0.017 | 1562 | 2987 | 1590 | 6139 | 2.46E+07 | 5.01E+08 | 5.23E+09 | 5.76E+09 |
| | Ground | 0.019 | 3452 | 3779 | 1477 | 8708 | 4.61E+07 | 5.80E+08 | 5.47E+09 | 6.10E+09 |
| | $R_{260m/ground}$ | 0.93 | 0.43 | 0.83 | 1.07 | 0.72 | 0.52 | 0.91 | 0.94 | 0.93 |
| non-Control Period | 260 m | 0.033 | 1296 | 3351 | 3469 | 8116 | 2.09E+07 | 6.41E+08 | 1.34E+10 | 1.41E+10 |
| | Ground | 0.033 | 3343 | 4386 | 3095 | 10824 | 4.57E+07 | 7.35E+08 | 1.38E+10 | 1.46E+10 |
| | $R_{260m/ground}$ | 0.93 | 0.39 | 0.82 | 1.06 | 0.76 | 0.46 | 0.92 | 0.93 | 0.92 |
| Clean | 260 m | 0.024 | 1328 | 3203 | 2475 | 7006 | 2.15E+07 | 5.88E+08 | 8.03E+09 | 8.64E+09 |
| | Ground | 0.027 | 3480 | 4338 | 2441 | 10258 | 4.70E+07 | 7.11E+08 | 8.83E+09 | 9.58E+09 |
| | $R_{260m/ground}$ | 0.87 | 0.37 | 0.78 | 0.98 | 0.68 | 0.44 | 0.86 | 0.89 | 0.88 |
| Polluted | 260 m | 0.054 | 1218 | 3702 | 5828 | 10748 | 1.94E+07 | 7.69E+08 | 2.61E+10 | 2.69E+10 |
| | Ground | 0.049 | 3022 | 4501 | 4633 | 12156 | 4.25E+07 | 7.93E+08 | 2.54E+10 | 2.63E+10 |
| | $R_{260m/ground}$ | 1.08 | 0.43 | 0.93 | 1.24 | 0.94 | 0.49 | 1.06 | 1.02 | 1.02 |

9) Figure 5: the data with higher time resolution, e.g. 10 min is recommended for figure a, b and c.

It is a good suggestion. We used hourly average mainly because the time resolution for the ground SMPS measurements was not constant in this study, for example, the time resolution for some periods is 30 min.

**Response to Reviewer #2**

This manuscript presents simultaneous measurements of particle number size distribution and particle chemical composition at a high level of 260 meter and at ground level. Information on comparison measurements in megacities such as Beijing can provide new insights into vertical distribution of particle formation and growth. In addition, comparison of measurements between control and non-control period will help to evaluate the effectiveness of the proposed emission control strategies. The paper in general is well written and should be publishable after some minor issues are address:

We thank the reviewer's positive comments.

1) The authors conclude that investigation of new particle formation and growth events at ground level in megacities needs to consider the influences of local cooking emissions. This is a very strong statement. Can the conclusion be generalized in megacities around the world or it is just constrained to some regions?

We thank the reviewer's comments. Cooking aerosols have been ubiquitously observed in megacities, and can contribute ~10 – 30 % of total OA, e.g., 16% in New York City (Sun et al., 2011), 19% in Fresno, CA (Ge et al., 2012) , 22-30% in London (Allan et al., 2010), 11-17% in Paris (Crippa et al., 2013) , 24% in Beijing (Huang et al., 2010), 24% in Lanzhou (Xu et al., 2014), and 24% in Hong Kong (Sun et al., 2016). The cooking contributions are even higher during the meal times. Therefore, the cooking emissions during the lunch time in the megacities can affect the particle growth in the daytime. Such an impact can drop rapidly from urban sites to rural areas due to the large decreases in cooking emissions (Ots et al., 2016) . Based on the results in previous studies and this work, we can draw such a conclusion, although it should be further explored in other megacities in the future studies.

2) The conclusion of emission controls enhancing new particle formation is another strong statement that may need a little more elaboration, for example, more in-depth data analysis and showing more evidences.

Thank the reviewer's comments. In this study, we found that the number of small particles during the control period was higher than those after the control period, while it was reversed for the accumulation mode particles. While emission controls was one of the reasons by reducing PM mass and suppressing particle growth, meteorological differences might be also important. As shown in Figure R1, the winds were dominantly from the north during the control period, while a large fraction was from the south after the control period. The prevailing northerly winds is one of the important causes for the low PM loadings during the control period, which is also one of the reasons leading to more frequent new particle formation events (Zhao et al., 2017). In the revised manuscript, we expanded the influences of meteorological factors on the differences of new particle formation between the two different peirods.

[Figure]

Figure R1.Wind rose plots (a) during the control period (20 August – 3 September) and (b) after the control period (4 September – 30 September), which are colored by wind speed (m s-1).

3) The authors claimed that "One of the major reasons is that emission controls decrease the gas precursors (e.g., SO₂ and NOₓ) and PM2.5 mass concentrations substantially and hence suppress the growth of particles to larger sizes." Is there something else that needs to be explored? For example, is it possible that accumulation mode particles are controlled from their direct emissions from industries or coal-fired power plants in addition to their contribution from secondary formation?

This is a good point. It is possible that the reductions of accumulation mode particles from direct emissions have played an important role. Unfortunately, we did not have measurements near industries and coal-fired power plants during this study. Modelling work might be helpful to address this important question in the future studies.

4) The authors used PMF to perform source apportionment and found two factors (factors 2&3) is likely associated with cooking emissions. The size distribution corresponding to factor 2 peaks at about 32 nm, which by and large falls within the small Aitken mode size range (15-40nm). This conclusion somehow is not consistent with the statements between L160-170 which attribute large Aitken mode particles (40-100 nm) to cooking emissions. In addition, while both factors 2 and 3 are attributed to local cooking emissions, what are the reasons that they are divided into two factors rather than combined into one?

We thank the reviewer's comments. Factor 3 was identified to be a factor mainly from cooking emission according to its diurnal variation and particle number size distribution, while factor 2 was a more complex factor that was not only associated with cooking emission but also contributed by particle growth. The size distributions of cooking-related factor 3 peaked at 50 and 60 nm at 260 m and ground level, respectively, and presented the dominant fractions between 40 – 100 nm, which is consistent with our conclusion at L160-170. Note that factor 2 was only resolved at ground level, which might indicate the different characteristics of cooking aerosols at different heights. For example, cooking aerosol at 260 m contains more large particles due to the condensation and/or coagulation processes during the transport from ground to high altitudes.

**Response to Reviewer #3**

This is an interesting work and this referee has a few minor comments for authors considering.

1) Lines 48-51, "The first continuous measurements of aerosol number size distributions within the city of Beijing indicated a high variability in number concentrations, and the variations were substantially different among dust storm, clean and polluted periods (Wehner et al., 50, 2004)." This is not fact, please double check and give a credit to a right one.

Thank the reviewer for pointing this out. We did find several earlier studies reporting the measurements of particle number size distributions in Beijing. We then revised this sentence as: "The continuous measurements of aerosols number size distributions from 3 nm to 10 μm within the city of Beijing in spring indicated a high variability in number concentrations, and the variations were substantially different among dust storm, clean and polluted periods (Wehner et al., 2004)".

2) Lines 56-57 "organics were found to be the dominant species in new particle formation events during the Beijing Olympic Games (Zhang et al., 2011)." No direct measurements for chemicals in <50 nm atmospheric particles were available in China, how can Zhang find organics to be the dominant species in new particle formation events? Argue?

We thank the reviewer's comment. This sentence was revised as "organics were found to be the dominant species of $PM_1$ during new particle formation events in summer in Beijing". The dominant species here refers to the bulk composition of $PM_1$. AMS can detect particles larger than 30 nm in aerodynamic diameter ($D_{va}$), while 50 nm in mobility diameter ($D_m$) is roughly equivalent to 70 nm ($D_{va}$) assuming spherical particles and a density of 1.4 g cm$^{-3}$. Therefore, the size-resolved AMS measurements in Zhang et al. (2011) can offer some insights into the composition of particles with $D_m$ < 50 nm although they were not analyzed and reported.

3) Lines 63-64 "Therefore, measurements of size-resolved number concentrations at high altitude with less local influences" Why? Local stacks at height can also greatly increase particle number

concentrations? If the sampling site is on the route for air plane landing or taking off, huge local emissions at height are also there.

We agree with the reviewer that local stacks at height can also greatly increase particle number concentrations, and if the sampling site is on the route for air plane landing or taking off, huge local emissions at heights are also there. The sentence "with less local influences" here means less traffic and cooking emissions from ground level. Following the reviewer's comments, we revised this sentence as: "Therefore, measurements of size-resolved number concentrations at high altitude with less local cooking and traffic influences are essential for elucidating the NPF and growth mechanisms"

4) Lines 84-92, SMPS suffers from a problem in accurately measuring particle size distribution in dynamic polluted air and is also unable to separate primary particles from grown new particles in size range > 30 nm. The weakness should be considered and mentioned.

We thank the reviewer's comment. In the revised manuscript, we added "According to previous comparisons of particle number size distributions between different SMPS or Differential Mobility Particle Sizers (DMPS), the measurement uncertainties between 20 and 200 nm can be ~10%, and even larger for particles outside this range (Wiedensohler et al., 2012)" so that the readers can know the uncertainties in comparisons of particle number size distributions between ground level and 260 m.

5) Lines 135-145, the referee has tested size distributions of particle number concentration and found that there was a dominant mode at ~20 nm. Of course, different cookings may not generate the same size distributions of particle number concentration. Please give more evidences for cooking source.

We thank the reviewer's comment. We chose two periods with significant cooking influences during this study, i.e., nighttime on 26 August and 1 September according to the PMF results in Zhao et al. (2017). As shown in Fig. R2, the average particle number size distributions of these two events were substantially different between 260 m and ground level. The number size distributions at ground

level was characterized by a single mode peaking at around 40 nm, which was similar to that from cooking activities (Buonanno et al., 2011). Comparatively, the particle number size distributions at 260 m were much broader and the concentrations were much lower than those observed at ground level.

[Figure]

Figure R2. The average particle number size distributions of two periods with substantially different on 26 August and 1 September at 260 m (dotted line) and ground level (line), respectively.

6) Lines 163-164 "Indeed, pronounced peaks for N 15-40 were often observed at evening time, likely indicating the influences of local emissions, e.g., cooking and traffic emissions." Yes, the two types of sources could be the cause. Vertical exchange of regional transported particles can also be a potential cause.

We agree with the reviewer's comment that the vertical exchange of regional transported particles can also be a potential cause. However, because of the relatively stable and low planetary boundary layer height at night, the vertical mixing is expected to be much weaker than daytime (Sun et al., 2015). In this study, the pronounced peaks for $N_{15-40}$ at 260 m were much lower than that at ground level, further indicating that local influences rather than vertical exchange were the major cause.

Because we cannot quantify and evaluate the impact of vertical change, we did not include such discussions in the manuscript.

7) Lines 175-195, changing size distributions of particle number concentration between two periods can also be due to the presence or absence of cloud-modification and should be considered. More clear days in control periods even strongly implied the possibility.

Right. We did observe substantially different size distributions between clean and polluted days, which were discussed in section 3.1. Similarly, more frequent new particle formation events during the control periods were associated with more clear days. In addition to regional emission control, we found that the prevailing northerly winds might have also played an important role. This is consistent with the reviewer's comment. In the revised manuscript, we slightly expanded the discussions on the influences of meteorological conditions during and after the control period.

8) Lines 220-221 "During the growth period, the GMD increased from 29 to 57 nm in 14 h at ground level, while it increased from 41 to 88 nm in 12 h at 260 m" It could be true, but hard to believe this. Please consider the weakness of SMPS measurements in dynamic urban atmospheres.

We thank the reviewer's suggestion. In this study, it is difficult for us to accurately evaluate the influences of the weakness of SMPS measurements on the particle growth. In fact, similar particle growth has been frequently observed in China, e.g., north China plain (Wang et al., 2013) and Shanghai (Xiao et al., 2015).

9) "Our results likely indicate that organics played an important role in the early stage of particle growth, while both organics and sulfate are important in the subsequent growth." Without direct measurements for chemicals in nucleation mode particles, it is really hard to say this. The same comment is applicable for lines 290-293.

Thank the reviewer's comment. We drew this conclusion mainly based the evolution of AMS $PM_1$ bulk composition. Because AMS/ACSM only detect particles with $D_{va}$ > 30 nm, our study mainly focus the growth of particles after 20 nm ($D_m$, which is approximately 30 nm in $D_{va}$). Therefore, the

changes in PM$_1$ bulk composition could indicate, at least partly, the different roles of aerosol species in the particle growth. We agree with the reviewer that accurate evaluation of the roles of aerosol species needs to measure the composition in nucleation mode particles (Smith et al., 2010).

10) Section 3.4, please consider cloud-modification for particle number size distribution.

We thank the reviewer's comment. The influences of meteorological conditions on the particle number size distributions were expanded in section 3.4.

**References**

Allan, J. D., Williams, P. I., Morgan, W. T., Martin, C. L., Flynn, M. J., Lee, J., Nemitz, E., Phillips, G. J., Gallagher, M. W., and Coe, H.: Contributions from transport, solid fuel burning and cooking to primary organic aerosols in two UK cities, Atmos. Chem. Phys., 10, 647-668, 10.5194/acp-10-647-2010, 2010.

Buonanno, G., Johnson, G., Morawska, L., and Stabile, L.: Volatility Characterization of Cooking-Generated Aerosol Particles, Aerosol Sci. Tech., 45, 1069-1077, 10.1080/02786826.2011.580797, 2011.

Crippa, M., DeCarlo, P. F., Slowik, J. G., Mohr, C., Heringa, M. F., Chirico, R., Poulain, L., Freutel, F., Sciare, J., Cozic, J., Di Marco, C. F., Elsasser, M., Nicolas, J. B., Marchand, N., Abidi, E., Wiedensohler, A., Drewnick, F., Schneider, J., Borrmann, S., Nemitz, E., Zimmermann, R., Jaffrezo, J. L., Prévôt, A. S. H., and Baltensperger, U.: Wintertime aerosol chemical composition and source apportionment of the organic fraction in the metropolitan area of Paris, Atmos. Chem. Phys., 13, 961-981, 10.5194/acp-13-961-2013, 2013.

Ge, X., Setyan, A., Sun, Y., and Zhang, Q.: Primary and secondary organic aerosols in Fresno, California during wintertime: Results from high resolution aerosol mass spectrometry, J. Geophys. Res.-Atmos., 117, D19301, 10.1029/2012jd018026, 2012.

Huang, X. F., He, L. Y., Hu, M., Canagaratna, M. R., Sun, Y., Zhang, Q., Zhu, T., Xue, L., Zeng, L. W., Liu, X. G., Zhang, Y. H., Jayne, J. T., Ng, N. L., and Worsnop, D. R.: Highly time-resolved chemical characterization of atmospheric submicron particles during 2008 Beijing Olympic Games using an Aerodyne High-Resolution Aerosol Mass Spectrometer, Atmos. Chem. Phys., 10, 8933-8945, 10.5194/acp-10-8933-2010, 2010.

Ots, R., Vieno, M., Allan, J. D., Reis, S., Nemitz, E., Young, D. E., Coe, H., Di Marco, C., Detournay, A., Mackenzie, I. A., Green, D. C., and Heal, M. R.: Model simulations of cooking organic aerosol (COA) over the UK using estimates of emissions based on measurements at two sites in London, Atmos. Chem. Phys., 16, 13773-13789, 10.5194/acp-16-13773-2016, 2016.

Smith, J. N., Barsanti, K. C., Friedli, H. R., Ehn, M., Kulmala, M., Collins, D. R., Scheckman, J. H., Williams, B. J., and McMurry, P. H.: Observations of aminium salts in atmospheric nanoparticles and possible climatic implications, P. Natl. Acad. Sci. USA., 107, 6634-6639, 10.1073/pnas.0912127107, 2010.

Sun, C., Lee, B. P., Huang, D., Jie Li, Y., Schurman, M. I., Louie, P. K. K., Luk, C., and Chan, C. K.: Continuous measurements at the urban roadside in an Asian megacity by Aerosol Chemical Speciation Monitor (ACSM): particulate matter characteristics during fall and winter seasons in Hong Kong, Atmos. Chem. Phys., 16, 1713-1728, 10.5194/acp-16-1713-2016, 2016.

Sun, Y., Du, W., Wang, Q., Zhang, Q., Chen, C., Chen, Y., Chen, Z., Fu, P., Wang, Z., Gao, Z., and Worsnop, D. R.:

Real-Time Characterization of Aerosol Particle Composition above the Urban Canopy in Beijing: Insights into the Interactions between the Atmospheric Boundary Layer and Aerosol Chemistry, Environ. Sci. Technol., 49, 11340-11347, 10.1021/acs.est.5b02373, 2015.

Sun, Y. L., Zhang, Q., Schwab, J. J., Demerjian, K. L., Chen, W. N., Bae, M. S., Hung, H. M., Hogrefe, O., Frank, B., Rattigan, O. V., and Lin, Y. C.: Characterization of the sources and processes of organic and inorganic aerosols in New York city with a high-resolution time-of-flight aerosol mass apectrometer, Atmos. Chem. Phys., 11, 1581-1602, 10.5194/acp-11-1581-2011, 2011.

Wang, Z. B., Hu, M., Sun, J. Y., Wu, Z. J., Yue, D. L., Shen, X. J., Zhang, Y. M., Pei, X. Y., Cheng, Y. F., and Wiedensohler, A.: Characteristics of regional new particle formation in urban and regional background environments in the North China Plain, Atmos. Chem. Phys., 13, 12495-12506, 10.5194/acp-13-12495-2013, 2013.

Wiedensohler, A., Birmili, W., Nowak, A., Sonntag, A., Weinhold, K., Merkel, M., Wehner, B., Tuch, T., Pfeifer, S., Fiebig, M., Fjäraa, A. M., Asmi, E., Sellegri, K., Depuy, R., Venzac, H., Villani, P., Laj, P., Aalto, P., Ogren, J. A., Swietlicki, E., Williams, P., Roldin, P., Quincey, P., Hüglin, C., Fierz-Schmidhauser, R., Gysel, M., Weingartner, E., Riccobono, F., Santos, S., Grüning, C., Faloon, K., Beddows, D., Harrison, R., Monahan, C., Jennings, S. G., O'Dowd, C. D., Marinoni, A., Horn, H. G., Keck, L., Jiang, J., Scheckman, J., McMurry, P. H., Deng, Z., Zhao, C. S., Moerman, M., Henzing, B., de Leeuw, G., Löschau, G., and Bastian, S.: Mobility particle size spectrometers: harmonization of technical standards and data structure to facilitate high quality long-term observations of atmospheric particle number size distributions, Atmos. Meas. Tech., 5, 657-685, 10.5194/amt-5-657-2012, 2012.

Xiao, S., Wang, M. Y., Yao, L., Kulmala, M., Zhou, B., Yang, X., Chen, J. M., Wang, D. F., Fu, Q. Y., Worsnop, D. R., and Wang, L.: Strong atmospheric new particle formation in winter in urban Shanghai, China, Atmos. Chem. Phys., 15, 1769-1781, 10.5194/acp-15-1769-2015, 2015.

Xu, J., Zhang, Q., Chen, M., Ge, X., Ren, J., and Qin, D.: Chemical composition, sources, and processes of urban aerosols during summertime in northwest China: insights from high-resolution aerosol mass spectrometry, Atmos. Chem. Phys., 14, 12593-12611, 10.5194/acp-14-12593-2014, 2014.

Zhao, J., Du, W., Zhang, Y., Wang, Q., Chen, C., Xu, W., Han, T., Wang, Y., Fu, P., Wang, Z., Li, Z., and Sun, Y.: Insights into aerosol chemistry during the 2015 China Victory Day parade: results from simultaneous measurements at ground level and 260 m in Beijing, Atmos. Chem. Phys., 17, 3215-3232, 10.5194/acp-17-3215-2017, 2017.